# Prevalence, risk factors, and consequences of hypothyroidism among pregnant women in the health region of Lleida: A cohort study

Júlia Siscart 1,2,3,4 *, Daniel Perejón 1,3,4,5, Maria Catalina Serna 1,3,5,6 *, Miriam Oros 1,3,4,5, Pere Godoy 6,7,8, Eduard Sole 9

1 Primary Care Research Institute IDIAP Jordi Gol, Catalan Institute of Health, Lleida, Spain, 2 Serós Health Center, Catalan Institute of Health, Lleida, Spain, 3 Family Medicine Department, University of Lleida, Lleida, Spain, 4 Therapeutic research group in primary care (GRETAP), Catalan Institute of Health, Lleida, Spain, 5 Eixample Health Center, Catalan Institute of Health, Lleida, Spain, 6 School of Medicine. Lleida University. Lleida. Spain Universitat de Lleida, Lleida, Spain, 7 Institut de Recerca Biomédica (IRBLleida), Lleida, Spain, 8 CIBER of Epidemiology and Public Health (CIBERESP), Instituto Carlos III, Madrid, Spain, 9 Pediatric Service of Hospital Arnau de Vilanova of Lleida, Lleida, Spain

* jvsiscart.lleida.ics@gencat.cat (JS); catalina.serna@udl.cat (MCS)

**Data Availability Statement:** The data used in this study are only available for the participating researchers, in accordance with current European and national laws. Thus, the distribution of the data

## Abstract

### Background

Primary maternal hypothyroidism is defined as the increase of TSH levels in serum during pregnancy. Hypothyroidism in pregnancy is the second most common endocrine disease, after diabetes mellitus, with a prevalence ranging between 3.2 and 5.5%. Its variability depends on ethnical differences. Hypothyroidism in pregnancy is associated with other chronic diseases and fetal and maternal outcomes.

### Objective

To analyze the prevalence of hypothyroidism among multiethnic pregnant women, and to evaluate the comorbidity with chronic diseases and outcomes leaded during pregnancy and newborn.

### Methods

Retrospective observational cohort study in pregnant women during the years 2012–2018 in the health region of Lleida. The relationship of hypothyroidism with different variables was analyzed by calculating the adjusted odds ratio (aOR) and the 95% confidence intervals (CI) with multivariate logistic regression models.

### Results

We analyzed a sample of 17177 pregnant women, which represents more than 92% of the total of pregnant women in the health region of Lleida. The annual prevalence of hypothyroidism was 5.7–7.1%. According to the region of origin, the lowest prevalence was found in the population from Sub Saharian Africa (2.1%), while the highest was from Asia and the Middle East (8.6%). Other factors associated with hypothyroidism were age, hypertension,

is not allowed. However, researchers from public institutions can request data from SIDIAP. Further information is available online https://www.sidiap.org/index.php/en/solicituds-en

**Funding:** "The authors declare support from PICARD 2021 subvention from Diputació de Lleida/IDIAP Jordi Gol and Institut Català de la Salut Lleida. The recipient is Júlia Siscart Viladegut and the grant number is 9F22/013. The funders had no role in study design, data collection and analysis, decision to publish, or preparation of the manuscript."

**Competing interests:** The authors have declared that no competing interests exist.

diabetes mellitus, and dyslipidemia. In addition, we did not observe an effect of hypothyroidism on the course of pregnancy, childbirth, and on the newborn. Finally, there was a good control of the disease during pregnancy.

## Conclusions

The prevalence of hypothyroidism in pregnancy was 6,5% in this study which depends on the country of origin, lower values were found in Sub Saharian African women and higher in those from Asia and the Middle East. Hypothyroidism was associated with age, diabetes mellitus, arterial hypertension, or dyslipidemia, and was not related to the Apgar score or the weight of the newborn.

## Introduction

Hypothyroidism is defined as a decreased production of thyroid hormone that may be due to primary damage on the thyroid gland or secondary as a result of hypothalamic or pituitary disease [1]. There are two types of hypothyroidism: frank, with a deficiency of thyroxine (T4) and the presence of symptoms; and subclinical, with normal T4 values and high thyrotropin values (TSH) and usually without symptoms [2].

During pregnancy, hormonal changes also affect the activity of the thyroid gland. First, glomerular filtration increases and so iodine clearance does it too. Iodine is an essential mineral in the metabolism of the thyroid hormones triiodothyronine (T3) and thyroxine (T4). Its reserves depend on its daily intake, thyroid catabolism, and its elimination via kidneys and its increased clearance is compensated by a higher production of T3 and T4 [3]. Second, the human chorionic gonadotropin hormone (hCG) increases. This glycoprotein shares the alpha subunit with TSH, the hypothalamic hormone that stimulates the pituitary gland and causes the thyroid gland to generate T3 and T4. Because of this similarity, hCG stimulates the hypothalamic-pituitary-thyroid axis, which produces an increase in T3 and T4 hormones in blood [4]. Finally, the increase of estrogen stimulates the production of thyroxine-binding globulin (TBG), which, when bound to T4, causes a deficiency of free T4 that the hypothalamus-pituitary-thyroid cycle tries to compensate for [5, 6].

Hypothyroidism is the second most prevalent endocrine disease after diabetes mellitus in pregnancy [7]. In different studies, the prevalence of the disease goes between 2 and 3% [8]: 0.2–0.5% for its clinical form and 0.25–5% for the subclinical one [9–11]. Ethnic differences have been described in TSH values and, therefore, in the prevalence of hypothyroidism [12].

In many studies, clinical and subclinical hypothyroidism has been associated with side effects in pregnancy, such as repeated miscarriage, preeclampsia or eclampsia, and gestational hypertension and diabetes. It has also been observed an increase of the risk of neonatal death, placental abruption, low weight birth, prematurity, fetal distress, intrauterine death, and intellectual deterioration [2, 13–16].

In this context, we aimed to analyze the prevalence of hypothyroidism among multiethnic pregnant women in the health region of Lleida and its association with chronic disease and outcomes during pregnancy. We decided to carry out this study because it is important for clinicians to have good epidemiological knowledge of the consequences and comorbidities prevalence on this disease and so they can act in certain way to prevent that.

**Table 1. Number of births registered in our sample in comparison to the Lleida health region per years.**

| Year | Deliveries in our sample | Deliveries from Idescat | Sample/Idescat |
|------|--------------------------|-------------------------|----------------|
| 2012 | 3635 | 3788 | 96% |
| 2013 | 3370 | 3535 | 95% |
| 2014 | 3308 | 3592 | 92% |
| 2015 | 3162 | 3426 | 92% |
| 2016 | 3180 | 3283 | 97% |
| 2017 | 3034 | 3197 | 95% |
| 2018 | 3001 | 3029 | 99% |

## Materials and methods

### Study design and data collection

We carried out a retrospective observational cohort study among pregnant women in the health region of Lleida during the years 2012–2018.

The data of women who had given birth at the Arnau de Vilanova Hospital between January 1st, 2012 and December 31st, 2018 were obtained through the CMBD database ("Conjunt Minim de Base de Dades"). In particular, the data of all the eligible patients assigned to a primary care unit derived from the computerized clinical history database E-CAP of the Catalan Health Institute; and data from Social Security prescriptions derived from the database of the Servei Català de Salut. In the Health Region of Lleida, a universal screening of thyroid function is carried out during the first trimester of pregnancy by determining the levels of TSH and T4.

This article is part of the Iler Pregnancy project, retrospective cohort study performed in Lleida with the aim of evaluating the prevalence of chronic diseases in pregnancy (hypothyroidism, depression, diabetes mellitus and obesity) and therapeutic adherence to prescribed drugs [17].

### Study population

Women who have had a birth at the Arnau de Vilanova University Hospital in Lleida between January 1st, 2012 and December 31st, 2018 were included in the study. Women who did not belong to Lleida health region were excluded. To evaluate the representativeness of the sample, we calculated the percentage of pregnant women studied compared to the total of pregnant women in the health region of Lleida. Data were obtained from the database of the "Instituto Statistics of Catalonia" (Idescat) (Table 1). We included data from the date of the last period to the date of delivery.

The following variables were recorded: presence of hypothyroidism, which corresponds to code E03.9 and E02 of the ICD-10; age; and levels of TSH and T4 in each trimester of pregnancy according to the laboratory reference values (Table 2). These hormonal levels were measured by immunoassay of enzyme chemiluminescence with the Beckman Coulter DXI 800 analyzer.

**Table 2. Reference values of TSH and T4 in each trimester of pregnancy according to laboratory criteria.**

| Trimester | TSH (nmol/L) | T4l (nmol/L) |
|-----------|--------------|--------------|
| First | 0.50–3.70 | 6.70–16.30 |
| Second | 0.31–4.35 | 5.80–13.90 |
| Third | 0.41–5.18 | 6.10–15.80 |

Other variables were also studied: region of origin (Sub Saharian Africa, Latin America, Asia and the Middle East, West Europe, Eastern Europe, and Maghreb) [18]; body mass index (BMI); number of pregnancy and twin pregnancy; abortions/miscarriages; prolonged or preterm delivery; cesarean section; diabetes mellitus (code O24.9 at CIE-10.); arterial hypertension (code I10-I16 at l'ICD-10); dyslipidemia (code E78 at l'ICD-10); depression (codes F32.0-F32.9, F33.0-F33.3, F33.8, F33.9, F34.1, or F41.2 at l'ICD-10); preeclampsia; risk during pregnancy (this classification is what is used in the real practice in Lleida Region Health, on the one hand we can categorised low risk and medium risk as "low risk". On the other hand we can categorised high risk and very high risk as "high risk); newborn weight, classified as low weight (<2500 gr), normal weight (2500–4000 gr), and macrosomia (>4000 gr) [19]; score of the Apgar test in the first minute and in the fifth, classified as good when $\geq$7 points and bad when <7 points [20]; and therapeutic adherence, we defined three levels of therapeutic adherence, as we observed in other studies: high, for patients who took more than 80% of the drug prescribed; medium, for those who took between 50 and 80%; and low, for those who took <50% [17]).

## Data analysis

We performed a descriptive analysis. Numerical variables were described by mean and standard deviation, and categorical variables by absolute and relative frequencies. Differences between groups were evaluated using Student's t-test or Chi-square test, depending on whether the variables were numerical or categorical, respectively. The relationship between hypothyroidism and other variables was analyzed using multivariate logistic regressions and adjusted Odds Ratio (OR) and the respective 95% confidence intervals (CI) were reported.

## Ethical aspects

This study was approved by the ethics and clinical research committee at the Institut d'Investigació IDIAP Jordi Gol under the code 19/195-P and carried out in accordance with the principles of the Declaration of Helsinki. Information was obtained from electronic medical records stored in the centralized ECAP database and extracted by the Department of Healthcare Evaluation and Research Management. Therefore, it was not necessary to ask participants for their informed consent. The variables in the ECAP database were processed anonymously and with full confidentiality guarantees as established by national law and Regulation 2016/679 of the European Parliament and of the Council on the protection of natural people with regard to the processing of personal data, and to the free movement of such data.

## Results

We started with a sample of 21375 women who had given birth at the Arnau de Vilanova Hospital in Lleida between 2012 and 2018 (both included). Of this sample 1625 women were excluded because they did not have a personal identification code (CIP), and 2573 because many data from the clinical history were missing. The final sample included were 17177 patients (Fig 1).

1127 women (6.5%) had a diagnosis of hypothyroidism, of whom 0.07% had clinical hypothyroidism, and 6.45% subclinical.

Every year the prevalence of hypothyroidism varied between 5.7% and 7.1% (Table 3), and patients showed a higher mean age in comparison to the rest: 31.7±5.7 years and 30.6±5.9 years, respectively (p<0.05). The mean BMI of patients was 25±5.2.

According to the region of origin, the highest percentages found were 71.4% from West European patients, 11.2% from Maghreb, 8.5% from Eastern Europe, and 5.1% from Latin

**Fig 1. Sample of pregnant women studied.**

America. A total of 402 pregnancies (39.5%) were classified as not at risk, 211 (20.7%) at medium risk, 370 (36.3%) at high risk, and a total of 36 (3.5%) at very high risk. 34 patients (3.9%) had abortions/miscarriages, 53 (6.2%) pre-term deliveries, 24 (2.8%) prolonged deliveries, and 751 (87.1%) delivered at term. Caesarean sections were performed in 189 patients (16.8%). Around 87% of the newborns from mothers either with or without hypothyroidism had normal weight, and no differences were observed in the percentages of newborns that were underweight or had macrosomia. Finally, no statistically significant differences were detected with the Apgar test at the first and fifth minutes between newborn from patients and from the rest of women (Table 4).

Thyroid hormone treatment was prescribed in 50.3% of the patients diagnosed with hypothyroidism. Among them, the mean adherence score was 79.6 ± 22.2, and 54% of these

**Table 3. Distribution of patients per year of pregnancy.**

| Year of delivery | Pregnant women with hypothyroidism (N: 1127) | Pregnant women studied (N: 17177) |
|---|---|---|
| 2012 | 158 (5.67%) | 2783 |
| 2013 | 173 (6.85%) | 2525 |
| 2014 | 161 (6.46%) | 2491 |
| 2015 | 166 (6.86%) | 2419 |
| 2016 | 170 (7.03%) | 2418 |
| 2017 | 165 (7.12%) | 2317 |
| 2018 | 134 (6.03%) | 2224 |

**Table 4. Characteristics of patients with hypothyroidism.**

| | Without hypothyroidism (N:16050) | With hypothyroidism (N: 1127) | P value |
|---|---|---|---|
| Age of the patients | 30.6 (SD 5.9) | 31.7 (SD5.7) | < 0.001 |
| Body mass index (BMI) | 24.9 (SD 4.9) | 25.0 (SD 5.2) | 0.364 |
| Region of origin | | | < 0.001 |
| Sub Saharian Africa | 822 (5.9%) | 18 (1.9%) | |
| Latin America | 668 (4.8%) | 49 (5.1%) | |
| Asia and the Middle East | 203 (1.5%) | 19 (1.9%) | |
| West Europe | 8768 (62.5%) | 693 (71.4%) | |
| Eastern Europe | 1451 (10.3%) | 82 (8.45%) | |
| Maghreb | 2124 (15.1%) | 109 (11.2%) | |
| Pregnancy number | | | 0.001 |
| 1 | 8472 (52.8%) | 537 (47.6%) | |
| 2 | 4802 (29.9%) | 379 (33.6%) | |
| 3 | 1725 (10.8%) | 145 (12.9%) | |
| 4 | 599 (3.7%) | 47 (4.2%) | |
| >4 | 452 (2.8%) | 19 (1.7%) | |
| Twin pregnancy | | | 0.721 |
| No | 16019 (99.8%) | 1126 (99.9%) | |
| Yes | 31 (0.2%) | 1 (0.1%) | |
| Pregnancy risk | | | <0.001 |
| Risk free | 7176 (50.1%) | 402 (39.5%) | |
| Medium risk | 4316 (30.1%) | 211 (20.7%) | |
| High risk | 2542 (17.8%) | 370 (36.3%) | |
| Very high risk | 280 (2.0%) | 36 (3.5%) | |
| Duration of the pregnancy | | | 0.867 |
| Abortion/miscarriage | 538 (4.4%) | 34 (3.9%) | |
| Pre-term | 716 (5.9%) | 53 (6.2%) | |
| Prolonged | 304 (2.5%) | 24 (2.8%) | |
| Full-term | 10545 (87.%) | 751 (87.1%) | |
| Cesarean | | | 0.639 |
| No | 13263 (82.6%) | 938 (83.2%) | |
| Yes | 2787 (17.4%) | 189 (16.8%) | |
| Newborn weight | | | 0.824 |
| Under weight | 850 (6.00%) | 60 (6.17%) | |
| Macrosomia | 959 (6.77%) | 61 (6.28%) | |
| Normal weight | 12352 (87.2%) | 851 (86.6%) | |
| Apgar test in the first minute | | | 1.000 |
| ≥ 7 | 13760 (97.5%) | 946 (97.5%) | |
| < 7 | 355 (2.5%) | 24 (2.5%) | |
| Apgar test at the fifth minute | | | 0.699 |
| ≥ 7 | 14006 (99.2%) | 964 (99.4%) | |
| < 7 | 111 (0.8%) | 6 (0.6%) | |

presented high adherence. Specifically, during the years of the study, from 2012 to 2018, the high adherence oscillate between 40.4–64.7% of the treated patients.

Fig 2 shows a multivariate analysis of the prevalence of hypothyroidism compared to other predictor variables. We observed that diabetes mellitus, hypertension, and dyslipidemia predisposed to the development of the disease. No statistically significant differences were found in terms of weight and depression between women with and without hypothyroidism.

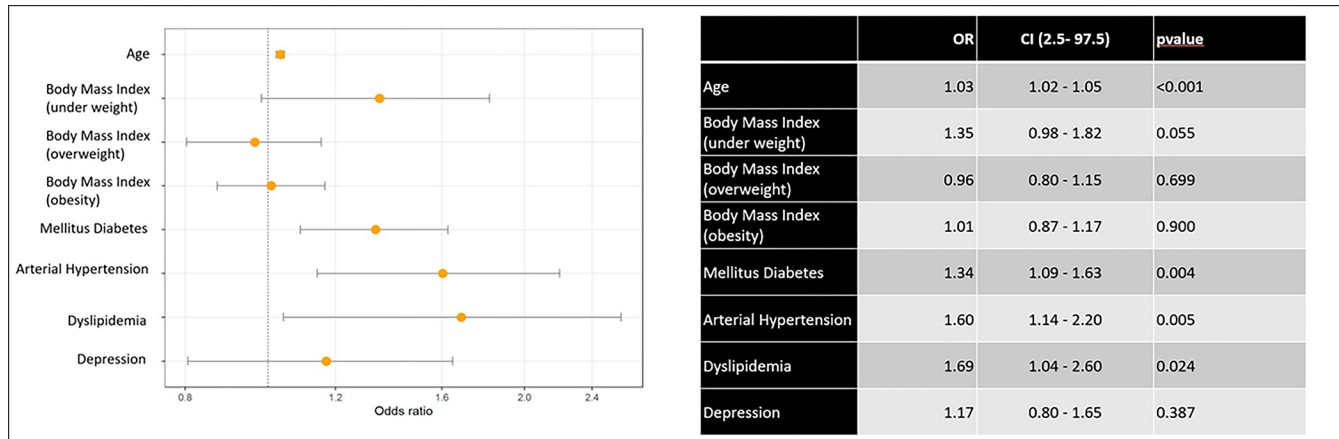

**Fig 2. Multivariate analysis of the prevalence of hypothyroidism and its association with other variables.**

Fig 3 shows a multivariate analysis of the prevalence of the association of maternal and fetal risk factors in women with hypothyroidism. We observed that hypothyroidism predisposed to a high and very high risk pregnancy. The rest of the variables did not present a statistically significant association with hypothyroidism.

## Discussion

We analyzed a sample of 17177 pregnant women representing the 92% of the total of pregnant women in the health region of Lleida. The prevalence of hypothyroidism was 6.5% and the mean age of the patients with hypothyroidism was significantly higher (31.7 ± 5.7 years) than the mean age of the rest (30.6 ± 5.9 years). The lowest prevalence was found in the population from Sub Saharian Africa (2.1%), while the highest is in those from Asia and the Middle East (8.6%). In this population the mean adherence score was 79.6 ± 22.2. Of these, 54% presented high adherence to treatment of hypothyroidism (18). Adherence to treatment and good control of TSH values in pregnancy may be related to the absence of adverse effects in the newborn. Other factors such as hypertension, diabetes mellitus, and dyslipidemia were found to be

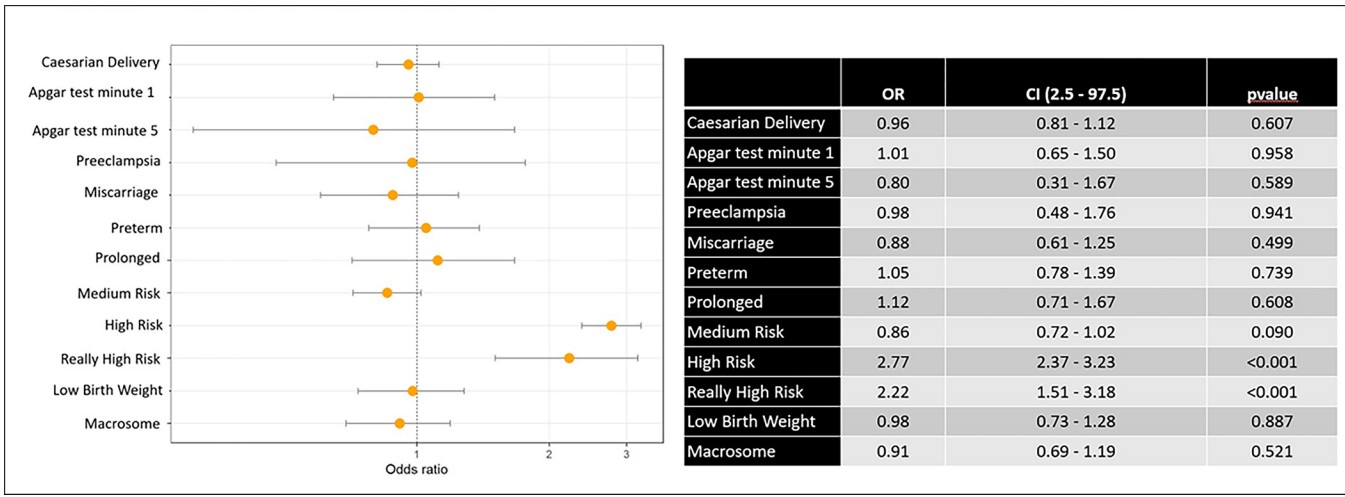

**Fig 3. Multivariate analysis of the association of maternal and fetal risk factors in women with hypothyroidism.**

related to hypothyroidism. Regarding the association of hypothyroidism with maternal and fetal outcomes, no differences were observed between patients and the rest of pregnant women.

The prevalence of hypothyroidism in our study was lower compared with the ones found by others such as: Nancy S. Pillai *et al.* [21], who reported in India a prevalence of 10.8%; Hatice Dulek *et al.* (8), who showed in Turkey values of prevalence at 8.9% for the subclinical form, and of 0.5% for the clinical; or L. Sletner *et al.* [12] describing in Norway a prevalence of 6.6% for subclinical hypothyroidism.

Age had already been described as a risk factor for developing hypothyroidism. Indeed, the "Guide to the diagnosis and treatment of thyroid diseases of endocrine metabolism" [22] advises screening for this disease from the age of 35; and the "Guide of the American Thyroid Association" [23] indicates being over 30 years of age as a risk factor. The prevalence of hypothyroidism has been associated with age in general population [24–26] and in pregnant woman as well [21].

The BMI of the patients without hypothyroidism was 24.9±4.92, while BMI of the patients with hypothyroidism was 25±5.24, these differences were not statistically significant. These results differ from the study conducted in India by Nancy S. Pillai et al. [21], as well as the one made by S.N. Ajmani et al. [27], where it was observed that the higher the BMI was associated with this disease. Furthermore, the study conducted by P. Gupta et al. [28] showed that high BMI were related with both hypothyroidism and hyperthyroidism.

Other variables significantly associated with hypothyroidism are arterial hypertension, diabetes mellitus, and dyslipidemia. Similar to our results, A.R Cappola *et al.* [29] and E. Berta *et al.* [30] and *Y. Han et al.* [31] related arterial hypertension to hypothyroidism. However, *G. Li et al.* [32] concludes that the incidence of arterial hypertension does not decrease with thyroid hormone replacement therapy.

Moreover, several studies have shown a relationship between diabetes mellitus and thyroid disorders [33, 34]. Finally, Dey *et al.* [35] described a relationship between hypothyroidism and dyslipidemia and concluded that good control of hypothyroidism favors good control of dyslipidemia.

Regarding the effects of the disease on the newborn, we did not observe differences in birth weight and Apgar Test during the first and fifth minutes. This can be caused by the early diagnosis and treatment of pregnant women with hypothyroidism, which results into good control of the disease, as we can see in the article of *G.Li et al.* [32]. Various studies, like the one by Barišić T *et al.* [36], also suggested that early detection and optimization of the treatment for hypothyroidism before and during the first trimester reduces the risk of adverse outcomes.

The differences observed in the prevalence of hypothyroidism according to ethnic groups, as well as its association with different comorbidities, lead us to plan a multidisciplinary public health strategy that helps mitigate the burden of this disease.

Future studies that analyze the possible influence of early detection and treatment of this chronic pathology, prior to and during pregnancy, will provide us with information regarding the benefits of good control of these diseases.

## Difficulties and limitations of the study

The strongest point of this study is the size of the sample that represents a high percentage of the population. Moreover, a universal screening has been performed in pregnant women, so it provides information on the majority of participants.

A possible limitation is the lack of data in some participants in our study, such as: women with two gestation periods during the same year, for example the end of the first pregnancy at

the beginning of the year and the beginning of the second pregnancy at the end of the same year, where the data could not have been separated; women for whom the Apgar Test and weight measurement of the newborn were not performed; and patients who carried out their follow-up in centers that do not belong to Social Insurance. It is estimated that these participants represent around 2.2% of the total in our health region; therefore, thanks to the universal coverage of the Spanish National Health System, it is unlikely that the results of the study have been affected by the missing data. Another limitation is represented by the fact that the presence of antithyroid antibodies was not analyzed because of the few existing data. Finally, we did not have available the data of socio-economical circumstances and environmental factors of the patients, that may also influence the results.

## Conclusions

The prevalence of hypothyroidism (6.5%) in pregnancy depends on ethnic origin. Sub Saharian African women presented lower hypothyroidism than those from Asia and the Middle East. A relationship of hypothyroidism with increasing age, hypertension, diabetes mellitus and dyslipidemia has been observed. However, no relation of hypothyroidism to the newborn's weight or Apgar score was appreciated. For all these reasons, we think that epidemiological knowledge of the disease by the of clinicians can help in early detection and treatment of hypothyroidism, as well as control of therapeutic adherence, can help to prevent its adverse effects in pregnancy and outcomes in offspring.

## Acknowledgments

The authors would like to acknowledge Dr. Miquel Butí for his valuable contribution and support to design and create the database, Joaquim Sol for his contribution to the statistics analysis, and the Gol i Gurina Foundation.

## Author Contributions

**Conceptualization:** Júlia Siscart, Daniel Perejón.

**Formal analysis:** Maria Catalina Serna, Miriam Oros.

**Investigation:** Júlia Siscart.

**Supervision:** Pere Godoy.

**Writing – original draft:** Júlia Siscart, Daniel Perejón.

**Writing – review & editing:** Maria Catalina Serna, Pere Godoy, Eduard Sole.

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
