## [Decision Letter · Decision Letter 0]

26 Dec 2022

PONE-D-22-30145Prevalence, risk factors, and consequences of hypothyroidism among pregnant women in the health region of Lleida: a cohort studyPLOS ONE

Dear Dr. Siscart,

Thank you for submitting your manuscript to PLOS ONE. After careful consideration, we feel that it has merit but does not fully meet PLOS ONE’s publication criteria as it currently stands. Therefore, we invite you to submit a revised version of the manuscript that addresses the points raised during the review process.

ACADEMIC EDITOR: Please revise the manuscript as suggested by the reviewer.

We look forward to receiving your revised manuscript.

Kind regards,

Mukhtiar Baig, Ph.D.

Academic Editor

PLOS ONE

Journal Requirements:

"The funders had no role in study design, data collection and analysis, decision to publish, or preparation of the manuscript"

5. PLOS requires an ORCID iD for the corresponding author in Editorial Manager on papers submitted after December 6th, 2016. Please ensure that you have an ORCID iD and that it is validated in Editorial Manager. To do this, go to ‘Update my Information’ (in the upper left-hand corner of the main menu), and click on the Fetch/Validate link next to the ORCID field. This will take you to the ORCID site and allow you to create a new iD or authenticate a pre-existing iD in Editorial Manager. Please see the following video for instructions on linking an ORCID iD to your Editorial Manager account: https://www.youtube.com/watch?v=_xcclfuvtxQ]

Reviewers' comments:

Reviewer's Responses to Questions

**Comments to the Author**

1. Is the manuscript technically sound, and do the data support the conclusions?

Reviewer #1: Yes

Reviewer #2: Partly

2. Has the statistical analysis been performed appropriately and rigorously? 

Reviewer #1: Yes

Reviewer #2: Yes

3. Have the authors made all data underlying the findings in their manuscript fully available?

Reviewer #1: Yes

Reviewer #2: Yes

4. Is the manuscript presented in an intelligible fashion and written in standard English?

Reviewer #1: Yes

Reviewer #2: Yes

5. Review Comments to the Author

Reviewer #1: The authors write in the limitation that socioeconomic status was not evaluated, however, it was part of the objectives. It should have been included.

Overall it is a well-written manuscript.

There are no concerns regarding dual publication, research ethics, or publication ethics.

Reviewer #2: Thank you for the opportunity to review the manuscript!

While the research topic is of importance to the clinicians working with pregnant women, the manuscript needs more details and some amendments to be brought to a standard format for publication.

Introduction:

- Please indicate at the end of the introduction why you have decided to conduct this study and what is the study significance and what gaps it aimed to address.

Materials and Methods

This section needs to address the following details:

- In the presented study, pregnancies were groups into low risk, medium risk, high risk and very high risk. Usually in real practice and for the sake of providing appropriate care to each group of women, pregnancies are categorised as “low risk” and “high risk”. Please indicate why you have chosen 4 gropes? Is this what is used in the real practice in the Lleida region to provide care to the women? If not, please clarify and re-group. Please modify the

Results accordingly.

- Please provide details of how you chose the criteria for adherence, how you calculated adherence score and what each score (lower/higher) means.

Results:

- “40.4-64.7% of the treated patients showed high adherence”. Why does adherence for one single group has a range? Please clarify or amend.

Table 4:

- “Duration of the pregnancy” has subgroups of “abortion”, “pre-term”, “prolonged”, and “full-term”. Please indicate the gestational age range for each item. Also, does the word “abortion” indicate both miscarriage and abortion, or does it refer to abortion only, or do you mean termination of pregnancy before 20th Weeks of gestation? They imply different meaning in clinical practice. Please clarify.

- “Newborn weight” has 3 sub-groups: underweight, macrosomia and normal weight. Please indicate the weight range for each group.

Figures 2 and 3:

- The standard format for presentation of regression results are “Adjusted OR”, “95% CI”, and “p-value”, each one sits on separate columns. Both lower and upper 95% CIs should be presented in one column for ease of reading. Please reformat the Figures.

Discussion

- “Difficulties and limitations of the study”: you have indicated that “… a lack of data for women with two gestation periods during the same year, where the data could not have been separated;”. Although not unlikely, it is uncommon that one woman gives birth to a preterm/term baby twice in one year. Do you mean both miscarriage/abortion and preterm/term pregnancies? Please clarify.

- What is the implications of the findings in practice? How can the readers use the results of the study to enhance their practice?

6. PLOS authors have the option to publish the peer review history of their article (what does this mean?). If published, this will include your full peer review and any attached files.

Reviewer #1: **Yes: **Dr Faiza Alam

Reviewer #2: No

---

## [Author Response · Author response to Decision Letter 0]

4 May 2023

Dear Editor,

Thank you for your review of the article. Your comments and those of the reviewers have allowed us to introduce some important improvements. I have made the modifications requested, and discuss them in the point-by-point response below.

Sincerely, 

Júlia Siscart Viladegut.

Reviewer 1

• Question (Q): The authors write in the limitation that socioeconomic status was not evaluated; however, it was part of the objectives. It should have been included. Overall it is a well-written manuscript. There are no concerns regarding dual publication, research ethics, or publication ethics.

• Answer (A): We have specified in the conclusions that we have not been able to provide the results due to lack of data.

“Finally, we did not have available the data of socio-economical circumstances and environmental factors of the patients, that may also influence the results.” (page 9, line 294-296)

Reviewer 2

• Q: Please indicate at the end of the introduction why you have decided to conduct this study and what is the study significance and what gaps it aimed to address.

• A: We have indicate what you suggest at the end of the introduction. 

“We decided to carry out this study because it is important for clinicians to have good epidemiological knowledge of the consequences and comorbidities prevalence on this disease and so they can act in certain way to prevent that.” (page 3, line 106-108)

• Q: In the presented study, pregnancies were groups into low risk, medium risk, high risk and very high risk. Usually in real practice and for the sake of providing appropriate care to each group of women, pregnancies are categorised as “low risk” and “high risk”. Please indicate why you have chosen 4 gropes? Is this what is used in the real practice in the Lleida region to provide care to the women? If not, please clarify and re-group. Please modify the results accordingly.

• A: This classification is what is used in the real practice in Lleida Region Health, on the one hand we can categorised low risk and medium risk as “low risk”. On the other hand we can categorised high risk and very high risk as “high risk”. 

• Q: Please provide details of how you chose the criteria for adherence, how you calculated adherence score and what each score (lower/higher) means.

• A: As we have observed in other studies we defined three levels of therapeutic adherence: high, for patients who took more than 80% of the drug prescribed; medium, for those who took between 50 and 80%; and low, for those who took <50%.

o Huber CA, Rapold R, Brüngger B, Reich O, Rosemann T. One-year adherence to oral antihyperglycemic medication and risk prediction of patient outcomes for adults with diabetes mellitus: An observational study. Medicine (Baltimore). 2016 Jun;95(26):e3994. doi: 10.1097/MD.0000000000003994. PMID: 27368004; PMCID: PMC4937918

o Hepp Z, Lage MJ, Espaillat R, Gossain VV. The association between adherence to levothyroxine and economic and clinical outcomes in patients with hypothyroidism in the US. J Med Econ. 2018 Sep;21(9):912-919. doi: 10.1080/13696998.2018.1484749. Epub 2018 Jun 22. PMID: 29865926

• Q: Results: “40.4-64.7% of the treated patients showed high adherence”. Why does adherence for one single group has a range? Please clarify or amend.

• A: We have clarified that. “Specifically, during the years of the study, from 2012 to 2018, the high adherence oscillate between 40.4-64.7% of the treated patients.” (page 6, lines 206-209)

• Q: - “Duration of the pregnancy” has subgroups of “abortion”, “pre-term”, “prolonged”, and “full-term”. Please indicate the gestational age range for each item. Also, does the word “abortion” indicate both miscarriage and abortion, or does it refer to abortion only, or do you mean termination of pregnancy before 20th Weeks of gestation? They imply different meaning in clinical practice. Please clarify.

• A: We have classified the duration of pregnancy as:

o “abortion”: Loss of baby before 20 weeks of gestation. 

o “pre-term”: Birth before 36 week of gestation.

o “prolonged”: Birth between 36-40 week of gestation.

o “full-term”: Birth before 40 week of gestation.

In addition, abortion and miscarriage both mean abortion.

• Q: “Newborn weight” has 3 sub-groups: underweight, macrosomia and normal weight. Please indicate the weight range for each group.

• A: We have classified the “Newborn weight” as: 

o “underweight”: weight less than 2500g at birth

o “normal weight”: weight between 2500 and 4000g at birth

o “macrosomia” weight more than 4000g at birth

• Q: Figures 2 and 3: - The standard format for presentation of regression results are “Adjusted OR”, “95% CI”, and “p-value”, each one sits on separate columns. Both lower and upper 95% CIs should be presented in one column for ease of reading. Please reformat the Figures.

• A: We have reformatted the figures, as you required us. 

• Q: Discussion: - “Difficulties and limitations of the study”: you have indicated that “… a lack of data for women with two gestation periods during the same year, where the data could not have been separated;”. Although not unlikely, it is uncommon that one woman gives birth to a preterm/term baby twice in one year. Do you mean both miscarriage/abortion and preterm/term pregnancies? Please clarify.

• A: We have clarified it at the discussion. “A possible limitation is the lack of data in some participants in our study, such as: women with two gestation periods during the same year, for example the end of the first pregnancy at the beginning of the year and the beginning of the second pregnancy at the end of the same year , where the data could not have been separated;” (page 8, lines 285-288)

• Q: What is the implications of the findings in practice? How can the readers use the results of the study to enhance their practice?

• A: The implications of the findings are explained at the end of conclusions, we have modified it a bit for its better understanding. “For all these reasons, we think that epidemiological knowledge of the disease by the of clinicians can help in early detection early detection and treatment of hypothyroidism, as well as control of therapeutic adherence, can help to prevent its adverse effects in pregnancy and outcomes in offspring” (page 8, lines 302-305)

---

## [Decision Letter · Decision Letter 1]

16 Aug 2023

PONE-D-22-30145R1Prevalence, risk factors, and consequences of hypothyroidism among pregnant women in the health region of Lleida: a cohort studyPLOS ONE

Dear Dr. Siscart,

Thank you for submitting your manuscript to PLOS ONE. After careful consideration, we feel that it has merit but does not fully meet PLOS ONE’s publication criteria as it currently stands. Therefore, we invite you to submit a revised version of the manuscript that addresses the points raised during the review process.

We look forward to receiving your revised manuscript.

Kind regards,

Mukhtiar Baig, Ph.D.

Academic Editor

PLOS ONE

Journal Requirements:

Reviewers' comments:

Reviewer's Responses to Questions

**Comments to the Author**

1. If the authors have adequately addressed your comments raised in a previous round of review and you feel that this manuscript is now acceptable for publication, you may indicate that here to bypass the “Comments to the Author” section, enter your conflict of interest statement in the “Confidential to Editor” section, and submit your "Accept" recommendation.

Reviewer #2: All comments have been addressed

Reviewer #3: (No Response)

2. Is the manuscript technically sound, and do the data support the conclusions?

Reviewer #2: Yes

Reviewer #3: Yes

3. Has the statistical analysis been performed appropriately and rigorously? 

Reviewer #2: Yes

Reviewer #3: Yes

4. Have the authors made all data underlying the findings in their manuscript fully available?

Reviewer #2: Yes

Reviewer #3: Yes

5. Is the manuscript presented in an intelligible fashion and written in standard English?

Reviewer #2: Yes

Reviewer #3: Yes

6. Review Comments to the Author

Reviewer #2: Many thanks for revising the manuscript!

Please add this description of risk groups to the text “This classification is what is used in the real practice in Lleida Region Health, on the one hand we can categorised low risk and medium risk as “low risk”. On the other hand we can categorised high risk and very high risk as “high risk”.

Please add details of adherence criteria to the text with the respective citations: “As we have observed in other studies we defined three levels of therapeutic adherence: high, for patients who took more than 80% of the drug prescribed; medium, for those who took between 50 and 80%; and low, for those who took <50%”.

The loss of pregnancy before 20th weeks can be abortion or miscarriage depending on the path. In the text, please replace “abortion” with “miscarriage/abortion”.

Figures 2 and 3: The usual format for presenting OR (CI) is up to two decimal points and for p-value is up to three decimal points. To maintain consistency, please revise Figures 2 and 3 accordingly.

Reviewer #3: The discussion section repeats most of the results already mentioned in the results section. It would be good if the discussion is re phrased to elucidate the implications of the findings from the study. Also address any discrepancies and offer insights into potential reasons for unexpected results. Suggestions for mitigation in clinical practice, counselling and future research endeavors should also be mentioned.

7. PLOS authors have the option to publish the peer review history of their article (what does this mean?). If published, this will include your full peer review and any attached files.

Reviewer #2: No

Reviewer #3: **Yes: **Syeda Sadia Fatima

---

## [Author Response · Author response to Decision Letter 1]

20 Sep 2023

Dear Editor,

Thank you for your review of the article. Your comments have allowed us to introduce some important improvements. I have made the modifications requested, and discuss them in the response below.

Sincerely, 

Júlia Siscart Viladegut.

Reviewer 2: Many thanks for revising the manuscript!

Question (Q):

Please add this description of risk groups to the text “This classification is what is used in the real practice in Lleida Region Health, on the one hand we can categorised low risk and medium risk as “low risk”. On the other hand we can categorised high risk and very high risk as “high risk”.

Answer (A):

I have added what you asked at methods.

Q:

Please add details of adherence criteria to the text with the respective citations: “As we have observed in other studies we defined three levels of therapeutic adherence: high, for patients who took more than 80% of the drug prescribed; medium, for those who took between 50 and 80%; and low, for those who took <50%”.

A: I add this details at methods.

Q:

The loss of pregnancy before 20th weeks can be abortion or miscarriage depending on the path. In the text, please replace “abortion” with “miscarriage/abortion”.

A: I have replaced the word abortin with miscarriage/abortion.

Q:

Figures 2 and 3: The usual format for presenting OR (CI) is up to two decimal points and for p-value is up to three decimal points. To maintain consistency, please revise Figures 2 and 3 accordingly.

A: Figures 2 and 3 have been revised and submitted again.

Reviewer 3: 

Q:

The discussion section repeats most of the results already mentioned in the results section. It would be good if the discussion is re phrased to elucidate the implications of the findings from the study. Also address any discrepancies and offer insights into potential reasons for unexpected results. Suggestions for mitigation in clinical practice, counselling and future research endeavors should also be mentioned.

A: I have modified some parts of the introduction of the discussion section, moreover I have expanded the end of the discussion section with what you suggested.

---

## [Editor Report · Decision Letter 2]

25 Sep 2023

Prevalence, risk factors, and consequences of hypothyroidism among pregnant women in the health region of Lleida: a cohort study

PONE-D-22-30145R2

Dear Dr. Siscart,

We’re pleased to inform you that your manuscript has been judged scientifically suitable for publication and will be formally accepted for publication once it meets all outstanding technical requirements.

Kind regards,

Mukhtiar Baig, Ph.D.

Academic Editor

PLOS ONE

---

## [Editor Report · Acceptance letter]

5 Oct 2023

PONE-D-22-30145R2 

Prevalence, risk factors, and consequences of hypothyroidism among pregnant women in the health region of Lleida: a cohort study 

Dear Dr. Siscart:

I'm pleased to inform you that your manuscript has been deemed suitable for publication in PLOS ONE. Congratulations! Your manuscript is now with our production department. 

Kind regards, 

on behalf of

Professor Mukhtiar Baig 

Academic Editor

PLOS ONE